# Neuroprotective Effect of Curcumin-Loaded RGD Peptide-PEGylated Nanoliposomes

**DOI:** 10.3390/pharmaceutics15122665

**Published:** 2023-11-24

**Authors:** Amina Ben Mihoub, Kamil Elkhoury, Janske Nel, Samir Acherar, Emilie Velot, Catherine Malaplate, Michel Linder, Shahrzad Latifi, Cyril Kahn, Marion Huguet, Frances T. Yen, Elmira Arab-Tehrany

**Affiliations:** 1LIBio Laboratory, University of Lorraine, F-54000 Nancy, France; benmihoubamina@hotmail.fr (A.B.M.); janske.nel@univ-lorraine.fr (J.N.); michel.linder@univ-lorraine.fr (M.L.); cyril.kahn@univ-lorraine.fr (C.K.); 2LCPM, CNRS, University of Lorraine, F-54000 Nancy, France; samir.acherar@univ-lorraine.fr; 3IMoPA, CNRS, University of Lorraine, F-54000 Nancy, France; emilie.velot@univ-lorraine.fr; 4UR AFPA Laboratory, Qualivie Team, University of Lorraine, F-54000 Nancy, France; catherine.malaplate-armand@univ-lorraine.fr (C.M.); marion.cizo@univ-lorraine.fr (M.H.); frances.yen-potin@inserm.fr (F.T.Y.); 5Department of Neuroscience, Rockefeller Neuroscience Institute, West Virginia University, Morgantown, WV 26506, USA; sl00092@hsc.wvu.edu; 6INSERM UMR_S 1116 DCAC Laboratory, University of Lorraine, F-54000 Nancy, France

**Keywords:** nanoliposomes, curcumin, PEG, targeting peptides, cortical neurons, neuroprotection

## Abstract

Curcumin is known for its anti-inflammatory, neuroprotective, and antioxidant properties, but its use in biological applications is hindered by its sensitivity to light, oxygen, and temperature. Furthermore, due to its low water solubility, curcumin has a poor pharmacokinetic profile and bioavailability. In this study, we evaluated the potential application of curcumin as a neuroprotective agent encapsulated in RGD peptide-PEGylated nanoliposomes developed from salmon-derived lecithin. Salmon lecithin, rich in polyunsaturated fatty acids, was used to formulate empty or curcumin-loaded nanoliposomes. Transmission electron microscopy, dynamic light scattering, and nanoparticle tracking analysis characterizations indicated that the marine-derived peptide-PEGylated nanoliposomes were spherical in shape, nanometric in size, and with an overall negative charge. Cytotoxicity tests of curcumin-loaded nanoliposomes revealed an improved tolerance of neurons to curcumin as compared to free curcumin. Wild-type SH-SY5Y were treated for 24 h with curcumin-loaded nanoliposomes, followed by 24 h incubation with conditioned media of SH-SY5Y expressing the Swedish mutation of APP containing a high ratio of Aβ40/42 peptides. Our results revealed significantly lower Aβ-induced cell toxicity in cells pre-treated with RGD peptide-PEGylated curcumin-loaded nanoliposomes, as compared to controls. Thus, our data highlight the potential use of salmon lecithin-derived RGD peptide PEGylated nanoliposomes for the efficient drug delivery of curcumin as a neuroprotective agent.

## 1. Introduction

Population aging is an irreversible global trend with major economic, socio-political, and health consequences. Scheltens et al. stated that one of the most undermining outcomes of aging is cognitive decline leading to dementia and neurodegenerative disorders [1]. Due to the current lack of effective pharmacological treatment and the irreversible damage that has already occurred by the time the first clinical signs appear, it is clear that preventive approaches are needed to slow down the physiological aging of the brain and the risk of neurodegenerative diseases such as Alzheimer’s disease [2,3]. During this long prodromal period, many biochemical changes occur in the brain leading to neuronal and synaptic dysfunction and subsequent cognitive impairment [4]. It is during this phase that preventive strategies, such as dietary modification and nutritional supplementation, may represent means towards reducing the progression of cognitive decline by favoring resistance of neurons to limit age-related cognitive decline, thereby reducing the risk of neurodegenerative diseases [5].

The brain contains high levels of lipids, which play multiple roles in preserving neuronal function and synaptic plasticity [6]. Among them, polyunsaturated fatty acids (PUFAs) are known to optimize synaptic membrane organization and function, rendering neurons more resistant to neurodegenerative processes [7,8]. As shown by Joffre et al., PUFA brain content is composed mainly of docosahexaenoic acid (DHA, n-3 PUFA) and arachidonic acid (AA, n-6 PUFA) [9]. DHA and AA are both critical for brain function and contribute to the prevention of age-related neurodegeneration and cognitive deficits [10,11]. The precursors of DHA and AA, α-linolenic and linoleic acids, respectively, are not synthesized in mammals, including humans, due to the lack of necessary enzymes. For these reasons, dietary intake of these PUFAs is essential.

Maintaining optimal levels of PUFAs in the aging brain requires developing means of delivery of these lipids. We recently reported the formulation of ω-3 PUFA-rich nanoliposomes (NL) from salmon head by-products [12]. Oral administration of these NL to mice led to an enrichment in PUFAs in their brains without significant weight gain or hepatic lipid accumulation, indicating the potential use of ω-3 PUFA-rich NL for delivery of PUFAs to the brain. Moreover, cell culture studies demonstrated the neurotrophic effects of NL in the primary culture of rat embryo neurons, which were able to internalize NL by a process resembling endocytosis [13]. By virtue of its properties, NL can be used to encapsulate both hydrophilic and hydrophobic molecules. Therefore, based on our previous findings, we propose that NL can be used as a bioactive vector since its internalization by neurons provides a means of delivery of the NL and its content directly into the cell.

Another challenge is to not only ensure brain-targeted delivery of NL but also to protect the integrity of NL during passage across the blood–brain barrier (BBB) [14]. As reported in the literature, polyethylene glycol (PEG) coating on the NL surface helps to improve the blood circulation time, tissue distribution, and physical stability of NL [15]. In addition, the RGD (arginine–glycine–aspartate) peptide is an extremely effective targeting agent. It is known to bind specifically to αvβ3 integrin receptors expressed in endothelial cells which are one of the components of the BBB. Thus, it is proposed that RGD coating on the NL surface can help to provide BBB targeting ability to NL [16]. We therefore decided to use these strategies (i.e., PEG and RGD coatings on the NL surface) to functionalize the ω-3 PUFA-rich NL derived from salmon fish, and to assess their potential neuroprotective properties.

As mentioned above, NL can be used as vectors to deliver neuroprotective agents. A potential molecule of interest is curcumin, which is a polyphenol that exhibits antioxidant, anti-inflammatory, and neuroprotective properties [17,18]. However, because curcumin has an extremely low water solubility, a large amount would need to be consumed to achieve any beneficial effect due to low bioavailability after ingestion [19]. Incorporation of sufficient amounts of this extremely hydrophobic molecule into food products has proven difficult, highlighting the importance of developing new ways to improve the water solubility and bioavailability of curcumin [20]. A successful solution, that has been tested by many groups, is the encapsulation of curcumin in NL formulations [21,22,23,24].

Based on all of the above, in this paper, we describe the synthesis and characterization of a PEG-peptide complex, which was then used to prepare RGD peptide-conjugated PEGylated ω-3 PUFA-rich NL loaded with curcumin. These NL formulations were then tested in cultures of neuronal cells exposed to media containing the β-amyloid peptide (Aβ) in order to evaluate their neuroprotective properties.

## 2. Materials and Methods

### 2.1. Materials

#### 2.1.1. Synthesis of c(RGDfC) Peptide and DSPE-PEG_5000_-c(RGDfC)

Ultrapure water (Mili-Q, r > 18 MΩ·cm) was used in all the experiments. Acetonitrile (ACN), acetic acid (AcOH), dicholoromethane (DCM), diethyl ether (Et_2_O), dimethylformamide (DMF), chloroform (CHCl_3_), and methanol (MeOH) were obtained from Sigma-Aldrich (Saint-Quentin Fallavier, France) and were used without further purification.

The synthesis of c(RGDfC) peptide required *N*,*N*,*N*′,*N*′-tetramethyl-*O*-(*1H*-benzotriazol-1-yl)uronium hexafluorophosphate (HBTU) (Iris Biotech GmbH, Germany) and acetic anhydride (Sigma-Aldrich, Saint-Quentin Fallavier, France), and *N*–methylmorpholine (NMM), *N*–methylpyrrolidinone (NMP), *N*,*N*-diisopropylethylamine (DIPEA), piperidine and tetrakis(triphenylphosphine)palladium(0) (Pd(PPh_3_)_4_) (Alfa Aesar, Haverhill, MA, USA) as reagents. Fmoc–*L*–Asp–OAll, Fmoc–Gly–OH, Fmoc–*L*–Arg(Pbf)–OH, Fmoc–*L*–Cys (Trt)–OH, Fmoc–*D*–Phe–OH and 2CTC resin (2-chlorotrityl chloride resin, 100–200 mesh, 1.0–2.0 mmol/g loading) were obtained from Iris Biotech GmbH (Marktredwitz, Germany). Trifluoroacetic acid (TFA) was purchased from Sigma-Aldrich (Saint-Quentin Fallavier, France) and triisopropylsilane (TIPS) from Alfa Aesar (Haverhill, MA, USA).

The synthesis of DSPE-PEG_5000_-c(RGDfC) required DSPE-PEG_5000_-Mal (≥95%, Biochempeg Scientific Inc., Watertown, MA, USA), and phosphate-buffered saline (pH 7.4) and 1 kDa dialysis membrane (Sigma-Aldrich, Saint-Quentin Fallavier, France).

#### 2.1.2. Materials for Nanoliposomes

Lecithin was extracted and purified from the heads of Salmo salar (Atlantic salmon) using a low-temperature enzymatic hydrolysis without any organic solvents [13]. Chloroform, methanol, and Dulbecco’s phosphate-buffered saline (DPBS) were purchased from Thermo Fisher Scientific (Ref: 14190144, Gibco, Massy, France). Curcumin with ≥94% (curcuminoid content) and ≥80% (Curcumin) (Ref: C7727) was purchased from Merck/Sigma-Aldrich, Saint-Quentin Fallavier, France).

### 2.2. Methods

#### 2.2.1. Synthesis of c(RGDfC) Peptide

The solid phase peptide synthesis (SPPS) of c(RGDfC) was performed on an automated ResPep XL peptide synthesizer (Intavis AG, Bioanalytical Instruments, Köln, Germany) and operated with a Multiple-parallel Peptide Synthesis Program at a 200 μmol synthesis scale. The iterative couplings of amino acids (AAs) were performed through a classical SPPS Fmoc/*t*Bu protocol with 2 equiv. of *N*-Fmoc-AA, 3 equiv. of HBTU as activating reagents, 9 equiv. of NMM, and 3 equiv. of NMP in a minimum of dry DMF (2 mL). The *C*-terminal extremity of aspartic acid (Asp, D) was protected by the allyl ester (OAll) group, and the side chains of arginine (Arg, R) and cysteine (Cys, C) were protected by 2,2,4,6,7-pentamethyldihydrobenzofuran-5-sulfonyl (Pbf) and trityl (Trt) groups, respectively. Fmoc deprotection was carried out using a solution of piperidine in dry DMF (20/80, *v*/*v*) and the unreacted AAs were capped with acetic anhydride at the end of each coupling phase. The OAll protecting group was removed using Pd(PPh_3_)_4_ (3 equiv) in a mixture of CHCl_3_*/*AcOH*/*NMM (37/2/1, *v*/*v*/*v*). Cleavage of the peptide from the resin with the removal of side chain protecting groups was achievable using a solution of TFA/TIPS/H_2_O (92.5/2.5/5, *v*/*v*/*v*).

The SPPS synthesis of c(RGDfC) is depicted in Appendix A. After attaching the side chain carboxyl group of Asp (2 equiv) to the 2CTC resin (1 equiv) in the presence of DIPEA (5 equiv) in DCM and then MeOH, the iterative couplings of AAs (Gly, *L*–Arg, *L*–Cys and *D*–Phe, respectively) were carried out. The *C*-terminal OAll and the *N*-terminal Fmoc protections were then removed successively before the cyclization through an intramolecular coupling. Finally, the peptide was cleaved from the resin with the removal of side chain protecting groups to yield the desired c(RGDfC). The peptide was precipitated in Et_2_O and purified by RP-HPLC using a Varian setup equipped with 2 Prostar 210 pimps, a reversed-phase Varian Pursuit column (5 µm, 21 mm × 150 mm), a Prostar 335 Varian PDA UV-visible detector, and a Prostar 363 Varian fluorescence detector. Data acquisition was performed by Varian Star Chromatography software. The UV detection was performed at 214 nm, and fluorescence detection at 274 nm (emission) after an excitation at 257 nm. The c(RGDfC) peptide was then obtained as a white solid in 42% yield using A/B elution (i.e., 10/90 for 15 min then 10/90 to 100/0 in 10 min and 100/0 to 10/90 in 5 min; A = acetonitrile + 0.1% TFA and B = water + 0.1% TFA).

The characterization of c(RGDfC) was accomplished by ^1^H NMR using a Bruker Advance 300 MHz spectrometer (Bruker, Wissenburg, Germany) in deuterated dimethyl sulfoxide (DMSO-*d*_6_) or acetic acid-*d*_4_ (CD_3_COOD) in deuterium oxide (D_2_O) (i.e., D_2_O/CD_3_COOD, 98/2, *v*/*v*) (Appendix A), but also by high-resolution mass spectrometry (HRMS) experiment using a microTOF Bruker (electrospray ionization ESI+, 50–1000 in low and 50–2500 in width) (Appendix A). The purity of c(RGDfC) was confirmed to be more than 98% by RP-HPLC under UV and fluorescence detections (Appendix A). The analysis by RP-HPLC was performed with the same equipment (see above) but with an Agilent Pursuit 5 C18 column (5 μm, 150 × 4.6 mm).

^1^H NMR (300 MHz, D_2_O/CD_3_COOD, 98/2, *v*/*v*) *δ*: 7.42–7.28 (m, 5H, H_Ar_, Phe), 4.75–4.68 (m, 2H, α-CH, Cys, and Asp), 4.38–4.32 (m, 1H, α-CH, Phe), 4.27–4.20 (m, 1H, α-CH, Arg), 4.23 (d, 1H, *J* = 15 Hz, α-CH, Gly), 3.53 (d, 1H, *J* = 15 Hz, α-CH, Gly), 3.26–3.18 (m, 2H, β-CH_2_, Cys), 3.06 (d, 2H, *J* = 8.1 Hz, δ-CH_2_, Arg), 2.92 (dd, 1H, *J* = 16.5 and 7.8 Hz, β-CH, Asp), 2.80–2.75 (m, 2H, β-CH_2_, Phe), 2.73 (dd, 1H, *J* = 16.5 and 7.8 Hz, β-CH, Asp), 1.95–1.83 (m, 1H, β-CH, Arg), 1.79–1.52 (m, 3H, β-CH and γ-CH_2_, Arg).

^1^H NMR (300 MHz, DMSO-*d*_6_) *δ*: 12, 25 (br s, 1H, CO_2_H, Asp), 8.29–8.24 (m, 1H, NH), 8.13 (dd, 2H, *J* = 7.5 and 2.5 Hz, NH), 7.91 (d, 2H, *J* = 8.4 Hz, NH), 7.46 (t, 1H, *J* = 5.7 Hz, NH), 7.28–7.18 (m, 5H, H_Ar_, Phe), 7.07 (br s, 2H, NH), 4,66–4.60 (m, 1H, α-CH), 4.53–4.47 (m, 1H, α-CH), 4.24–4.03 (m, 3H, α-CH), 3.36–2.95 (m, 4H, α-CH, Gly, β-CH_2_, Cys and δ-CH, Arg), 2.84–2.16 (m, 5H, δ-CH, Arg, β-CH_2_, Asp and β-CH_2_, Phe), 1.76–1.84 (m, 1H, β-CH, Arg), 1.60–1.40 (m, 3H, β-CH and γ-CH_2_, Arg, β-CH, Asp).

HRMS (ESI) calculated for C_24_H_35_N_8_O_7_S [M+H]^+^ *m/z* 579.2349, found 579.2337.

RP-HPLC (C18 column) retention time (t_R_): 6.842 mn (UV detection, λ_abs_ = 214 nm) and 6.952 mn (fluorescence detection, λ_abs_ = 257 nm and λ_em_ = 274 nm).

#### 2.2.2. Synthesis of DSPE-PEG_5000_-c(RGDfC)

The conjugation of c(RGDfC) to DSPE-PEG_5000_-Mal was achieved using a Thiol–Maleimide Click Chemistry (TMCC) reaction (Figure 1a). Briefly, c(RGDfC) (1.2 equiv.) was dissolved in a minimum volume of DMF and diluted with PBS (pH = 7.4). A solution of DSPE-PEG_5000_-Mal (1 equiv.) in DMF was added dropwise in five portions to the solution of c(RGDfC), resulting in a final DMF/PBS volume ratio of 1/1 in the mixture. The mixture was stirred overnight at room temperature. The reaction was monitored by ^1^H NMR (300 MHz in DMSO-*d*_6_) until the disappearance of the maleimide double bond protons at 6.69 ppm (Figure 1b). The excessive c(RGDfC) peptide was removed by dialysis (MWCO 1 kDa). The DSPE-PEG_5000_-c(RGDfC) (85% yield) lot was confirmed by MALDI-TOF (Figure 1c) using an Axima Performance MALDI-ToF system (Shimadzu Biotech, Long Beach, CA, USA) equipped with a nitrogen laser (λ = 337 nm) and 2,5-dihydroxybenzoic acid as matrix (0.5 µL of matrix solution mixed with 0.5 µL of sample solution). Mass spectra of DSPE-PEG_5000_-Mal and DSPE-PEG_5000_-c(RGDfC) were acquired in a linear mode in the mass range 0–10,000 Da, with the following parameters: power, 100; profiles, 50; shots per profile, 10; pulse rate, 10 Hz; pulsed extraction optimized at 3000 Da; raster type, regular annular.

#### 2.2.3. Preparation of Nanoliposomes

Two formulations of nanoliposomes (NL) were prepared for this study, namely the empty NL containing no drug or compound, and curcumin-loaded nanoliposomes (NLC) in which curcumin was encapsulated in the lipid bilayer of the liposomes. Both the NL and NLC were prepared either with or without PEG coating (i.e., NLP or NLPC, respectively) and RGD peptide coating (i.e., NLPP or NLPPC, respectively). NL were prepared using the thin film rehydration method to yield homogenous monodisperse liposomes. For the synthesis of empty NL without RGD peptide or PEG, 50 mg of salmon lecithin was dissolved in 1.5 mL of chloroform/methanol (1:2, *v*/*v*), and a thin lipid film was formed on the wall of the flask using a Rotavapor by completely evaporating the solvent under vacuum at 50 °C. The thin film was hydrated with 2.5 mL of sterile cell culture DPBS (thus yielding a 2% (*w*/*v*) salmon lecithin NL formulation) and incubated at room temperature with gentle stirring overnight (approx. 16 h). Following the overnight rehydration, the solution was sonicated (Vibra-Cell 75115 Sonicator, 500 Watt, Bioblock Scientific Co., Illkirch, France) for 4 min (pulse of 1 s on and 1 s off, thus 8 min total) at 40 kHz at 40% power in an ice bath. For the preparation of other formulations, 15 mg of curcumin and/or 8 mg of PEG-peptide was dissolved along with the salmon lecithin in chloroform/methanol before thin film production. To prevent lipid oxidation and curcumin degradation, all formulations were flushed with nitrogen (N_2_ gas) and protected from light throughout the preparation steps. Samples were stored in glass bottles under nitrogen and in the dark at 4 °C.

#### 2.2.4. Physicochemical Characterization

The mean size and ζ-potential of NL were measured by dynamic light scattering (DLS) using a Malvern Zetasizer Nano ZS (Malvern Instruments Ltd., Malvern, UK). The samples were diluted (1:200) with ultrapure distilled water and filtered with sterile 0.45 µm Supor^®^ membrane syringe filters to remove excess salmon lecithin, curcumin, and/or PEG-peptide, and then with sterile 0.22 µm Supor^®^ membrane syringe filters before use in cell culture studies. Size measurements were performed at 25 °C with an absorbance of 0.01, a fixed scattering angle of 173°, and a refractive index of 1.471 at 25 °C in standard cells. ζ-potential measurements to determine the surface net charge of the formulations were performed under similar conditions in cells equipped with gold electrodes. The sizes are shown as the z-average mean for the liposomal hydrodynamic diameter (nm). PDI represents the width of vesicle size distribution ranging from 0 to 1, of which PDI ≤ 0.1 is considered to be highly monodispersed, whilst values of 0.1 to 0.4 and >0.4 are considered to be moderately and highly polydispersed, respectively. Three independent DLS measurements were performed for each NL type immediately after sonication.

#### 2.2.5. Morphology

Transmission electron microscopy (TEM) was used to observe the morphology and microstructure of NL and NLPPC using the negative staining method. Briefly, samples were diluted with ultrapure distilled water (25 fold), mixed 1:1 with an aqueous solution of ammonium molybdate (2%), and incubated for 3 min at room temperature. A drop of this solution was placed on a Formvar–carbon-coated copper grid (200 mesh, 3 mm diameter HF 36) for 5 min. After drying, micrographs were taken using a Philips CM20 TEM operating at 200 kV and recorded using an Olympus TEM CCD camera.

#### 2.2.6. Encapsulation Efficiency of Curcumin

The percentage of curcumin encapsulated within the salmon lecithin NL was determined by a UV-1800 spectrophotometer (Shimadzu, Noisiel, France). Briefly, freshly prepared NLC with or without PEG-peptide were centrifuged at 9000× *g* for 15 min to separate the unloaded curcumin crystals from the nanoliposomes. The sedimented curcumin was diluted in methanol and measured at 425 nm by UV spectrometry. Similarly, the curcumin in the supernatant of the centrifuged sample (i.e., NLC with or without PEG-peptide) was measured. The content of curcumin was then calculated from a calibration curve. The experiments were performed in triplicate, and the encapsulation efficiency (EE) was calculated as
EE % = ((total drug − free drug)/total drug) × 100

#### 2.2.7. Cell Culture

Primary cultures of cortical neurons from rat embryos were prepared as described previously [25]. Cells were maintained in serum-free M2 neuronal culture medium (Invitrogen, Illkirch, France) containing 0.5 µM insulin, 60 µM putrescine, 30 nM sodium selenite, 100 µM transferrin, 10 nM progesterone, and 0.1% (*w*/*v*) ovalbumin, all factors obtained from Sigma-Merck. Cell cultures were maintained at 35 °C in a humidified 6% CO_2_ atmosphere. Day 0 was the day on which cells were isolated and plated. Curcumin in DMSO or nanoliposomes was added on Day 3, followed by the analysis of cell viability 24 h on Day 4 using the MTT method [25]. Control values were considered as 100% viability.

The neuronal cell line SH-SY5Y (wild-type, WT), or the same cell line expressing the Swedish mutation of APP (APP, [25]) were kind gifts from Marie-Claude Potier. Cells were grown and maintained at 37 °C in a 95% air/5% CO_2_ environment in high-glucose DMEM (Gibco Ref: 11960044) containing 10% heat-inactivated fetal bovine serum (FBS) and 1% penicillin–streptomycin (Invitrogen, Ref: P4333).

Cell culture experiments were performed using the Bioavailability–Bioactivity (Bio-DA) platform.

#### 2.2.8. Test of Neuroprotection

SH-SY5Y WT or APP cells at 80% confluence were differentiated into neurons by treatment with 10 µM retinoic acid (Invitrogen Ref: R2625) in a Neurobasal medium (Gibco, Ref: 21103049) containing 3% FBS and 1% penicillin–streptomycin. After 4 days, differentiated WT cells were washed in PBS, and then incubated 24 h with the different nanoliposome formulations. Differentiated APP cells were washed in PBS, and then incubated 24 h with serum-free neurobasal medium. The conditioned media (CM) was recovered and then centrifuged at 2000× *g* for 10 min to eliminate any debris. The Aβ peptide levels in CM were measured using the selective solid-phase sandwich ELISA kits to detect human Aβ40 (residues 1 to 40) and Aβ42 (residues 1 to 42) full-length peptides (Invitrogen).

The CM was diluted at a ratio of 1:1 with neurobasal medium and then added to the differentiated WT cells that had been preincubated with the nanoliposome preparations. After 24 h incubation at 37 °C, cell viability was measured using MTT. To assess apoptosis, cell nuclei were visualized using 4,6-diamidino-2-phenylindole (DAPI, 1:10,000, Molecular Probes, Eugene, OR, USA). The cells, grown on a glass coverslip, were washed in PBS, incubated at room temperature for 10 min with DAPI, and then washed with PBS. Images were acquired using an inverted confocal microscope (Fluoview10, Olympus). To evaluate the percentage of apoptotic cells, five independent fields of microscope were counted (around 50 cells) in three separate experiments. The number of apoptotic nuclei is shown as mean ± SD.

#### 2.2.9. Statistical Analyses

Results were expressed as means ± SD. For the cell culture studies, the values for each set of cells treated with nanoparticles were calculated as % of control cells incubated in the absence of nanoparticles from the same set of cells. The Kolmogorov–Smirnov test was used to verify normal distribution, and statistical analysis was performed using ANOVA; statistical significance was considered as *p* < 0.05.

## 3. Results and Discussion

### 3.1. Synthesis and Characterization of Targeted Peptide c(RGDfC)

The targeted cyclopeptide c(RGDfC) was prepared using Fmoc/tBu solid-phase peptide synthesis (SPPS) on the 2-chlorotrityl chloride (2CTC) resin (Appendix A). The use of Fmoc–*L*–Asp–OAll and the attachment to the resin through its side chain carboxyl group allowed cyclization to be carried out on the resin. In the end, the simultaneous cleavage under acidic conditions of the cyclopeptide from the resin and of the lateral chains, followed by reversed-phase high-performance liquid chromatography (RP-HPLC) purification, afforded c(RGDfC) as a white solid in 42% overall yield and a purity of more than 98%. The c(RGDfC) peptide was characterized by ^1^H nuclear magnetic resonance (^1^H NMR, Appendix A), high-resolution mass spectrometry (HRMS, Appendix A), and RP-HPLC (Appendix A).

### 3.2. Synthesis and Characterization of RGD Peptide-PEGylated DSPE (DSPE-PEG_5000_-c(RGDfC))

DSPE-PEG_5000_-c(RGDfC) was synthesized by conjugation of the cysteine residue of c(RGDfC) peptide and the maleimide unit of the commercially available DSPE-PEG_5000_-Mal through a Thiol–Maleimide Click Chemistry (TMCC) reaction in a mixture of DMF/PBS (pH 7.4) (1/1, *v*/*v*) (Figure 1a). The monitoring of the reaction was performed using ^1^H-NMR spectroscopy (300 MHz, DMSO-*d*_6_) (Figure 1b). The majority of the new product’s NMR peaks overlapped with those of the starting PEGylated DSPE. However, some characteristic peaks of c(RGDfC) and DSPE-PEG_5000_-Mal were still visible in the ^1^H NMR spectrum of the conjugated DSPE-PEG_5000_-c(RGDfC) (i.e., red and green boxes in Figure 1b) and, more importantly, the disappearance of the maleimide alkene proton peaks at 6.69 ppm unambiguously indicated the success of the TMCC reaction (Figure 1b). The final yield of targeted DSPE-PEG_5000_-c(RGDfC) was 85% after dialysis removal of the residual c(RGDfC) (MWCO 1 kDa). MALDI-TOF mass spectrometry was used to confirm the obtention of the desired DSPE-PEG_5000_-c(RGDfC) product from simple molecular weight determination (Figure 1c). Indeed, the MALDI-TOF results showed a molecular weight of 577.42 for c(RGDfC) (Figure 1c, top) and an average molecular weight of 4580.36 for DSPE-PEG_5000_-Mal (Figure 1c, middle). This average molecular weight shifted to 5157.42 for DSPE-PEG_5000_-c(RGDfC) (Figure 1c, down), revealing the success of the TMCC reaction.

### 3.3. Preparation and Characterization of Different Nanovesicles

NL formulations were prepared using the thin film method to achieve a high entrapment efficiency of curcumin, followed by sonication to achieve a homogenous size distribution of nanoparticles. Immediately after sonication, the average hydrodynamic size, polydispersity index (PDI), and ζ-potential of the empty NL and curcumin-loaded NL either with or without PEG coating (NLPC or NLC, respectively) and c(RGDfC) coating (NLPP and NLPPC) were measured. The mean particle size for empty NL was 78.67 ± 0.97 nm, which increased to 107.73 ± 1.17 nm for the NLP and 127.2 ± 4.52 nm for the NLPP (Figure 2a). The nanoliposomes presented with a narrow distribution with a PDI < 0.3 (Figure 2b), after both a 0.45 and 0.22 µm filtration to sterilize the sample for use in cell culture. All formulations presented a negative ζ-potential of −39.53 ± 0.78 mV for NL, −30.37 ± 0.63 mV for NLP, and −26.23 ± 0.72 mV for NLPP (Figure 2c), consistent for lecithin-derived liposomes [26].

Notably, a low ζ-potential value indicates that the nanoparticle dispersion presents good stability as the greater the ζ-potential magnitude, the greater the repulsion between particles which leads to a more stable colloidal dispersion [27]. Furthermore, the increases in size and ζ-potential of NL after PEG and peptide coatings were indicative of the successful PEG and peptide modifications on the NL surface [28,29].

Likewise, the encapsulation of curcumin caused a change in size, PDI, and ζ-potential in the liposomes; NLC were 131.90 ± 1.79 nm in size, with a PDI of 0.23 ± 0.01 and a ζ-potential of −37.13 ± 0.54 mV (Figure 2a–c). With the addition of PEG on the NL surface, the size and ζ-potential of the NLPC increased to 144.80 ± 1.34 nm and −23.63 ± 0.82 mV, while the PDI remained largely similar at 0.22 ± 0.01 (Figure 2a–c). Also, with the addition of peptide-PEG on the NL surface, the size and ζ-potential of the NLPPC increased to 128 ± 2.45 nm and −23.9 ± 0.79 mV, whilst the PDI remained largely similar at 0.24 ± 0.03 (Figure 2a–c).

The encapsulation efficiencies of curcumin in different salmon lecithin liposomes formulations (2%, *w*/*v*) were 94.7± 3.3% for NLC, 95.7 ± 2.1% for NLPC, and 98.4 ± 1.4% for NLPPC, with no significant difference between the three different formulations (Figure 2d). As curcumin is a highly unstable molecule, being degraded by light, pH, and temperature, its encapsulation into drug carriers such as liposomes is essential to retain its physiochemical properties and stability. Furthermore, transmission electron microscopy (TEM) images showed the presence of spherical nanovesicles in both NL and NLPPC formulations (Figure 2e,f).

### 3.4. Biocompatibility of Curcumin-Loaded Nanoliposomes (NLC)

We previously showed the biocompatibility of empty salmon lecithin NL with cultured neurons, establishing no significant loss of viability with 10 µg/mL NL [12]. Here, we compared the biocompatibility of curcumin alone with that of NLC. Primary cultures of rat embryo cortical neurons were incubated on day 3 after culture for 24 h at 37 °C with increasing concentrations of curcumin alone or encapsulated into NL. Identical concentrations of NLC were added based on the amount of salmon lecithin in the preparation, prepared with the indicated concentrations of curcumin. No significant decrease in cell viability was observed with NLC, in contrast to the cytotoxicity of curcumin which was concentration dependent (Figure 3). Although known for its neuroprotective properties, curcumin can become toxic above a certain concentration, which may vary according to the cell model used. In neuronal models, toxicity has been described after exposure to 20 µM for at least 24 h [30,31,32,33]. In the model used here, of primary rat cortical neurons, doses of 10 µM and 20 µM for 24 h resulted in a decrease on the order of 30% and 50%, respectively (Figure 3). On the other hand, no significant loss of viability was observed in cells incubated with NLC. Therefore, encapsulation of curcumin in NL significantly reduced cytotoxic effects as compared to free curcumin at similar concentrations.

### 3.5. Neuroprotective Effects of PEGylated and Peptide PEGylated Nanoliposomes (NLP and NLPP)

We next investigated the ability of the different functionalized NL to reduce Aβ-induced toxicity. First, we determined that the different functionalized empty (NL, NLP, NLPP) or curcumin-encapsulated NL (NLC, NLPC, NLPPC) were not toxic. Differentiated SH-SY5Y neurons were incubated for 24 h at 37 °C with 10 µg/mL of the NL formulations, followed by measurement of cell viability using MTT. Results indicated no significant changes in cell viability as compared to control cells (Figure 4a, CTL). This is consistent with previous results using empty NL and primary cultures of rat embryo cortical neurons (8) and confirms that the different formulations of NL, either empty or loaded with curcumin, were not cytotoxic at these concentrations. Next, differentiated SH-SY5Y neurons were pretreated for 24 h at 37 °C with the different NL formulations that were empty or encapsulating curcumin (20 µM). Conditioned media (CM) containing Aβ peptide (Aβ 1-40: 568 ± 120.2 pg/mL and Aβ 1-42 41.3 ± 13.6 pg/mL, n = 6) from differentiated SH-SY5Y APP neurons were then added to the cells. After 24 h incubation at 37 °C, cell viability was measured using MTT. Incubation of neurons with CM alone led to a significant decrease in cell viability as compared to control cells (Figure 4b–d, CTL vs. CM alone). When compared to control cells (CTL), CM-induced decrease in cell viability remained significant, but to a lesser degree, if cells were preincubated with the different empty NL (Figure 4b, NL + CM; Figure 4c, NLP + CM, Figure 4d, NLPP + CM). When cells were preincubated with curcumin-loaded NL formulations (Figure 4b, NLC + CM, Figure 4c, NLPC + CM, Figure 4d, NLPPC + CM), and then treated with CM, cell viabilities were no longer significantly different from those of the corresponding control cells (CTL). In addition, for NLC (Figure 4b) and NLPC (Figure 4c), the cell viability after CM treatment was statistically significantly higher as compared to NL or NLP formulation in the absence of curcumin, whereas that for NLPPC did not achieve statistical significance. These results thus demonstrated the feasibility of delivering sufficient levels of curcumin for neuroprotection without cytotoxic effects.

To assess the level of apoptosis, a series of cells plated on slides were incubated in a similar manner as described above, labeled with DAPI, and the number of fragmented nuclei was counted. CM treatment led to a significant increase in apoptotic nuclei (Figure 5a–c, CTL vs. CM alone), confirming the pro-apoptotic effects of the Aβ-containing CM (22). For both NL (Figure 5a, NL + CM) and NLP (Figure 5b, NLP + CM), there was no statistically significant change in the number of apoptotic nuclei after CM treatment (Figure 5a,b, CM alone). However, the number of apoptotic nuclei in CM-treated cells preincubated with curcumin-loaded NLPC (Figure 5b, NLPC + CM), empty NLPP, or curcumin-loaded NLPPC (Figure 5c, NLPP + CM, NLPPC + CM) were not significantly different from those of control cells (Figure 5c,d, CTL).

Curcumin-loaded NLC significantly reduced cytotoxic effects as compared to curcumin alone at similar concentrations, thereby improving the biocompatibility of curcumin with neurons. Furthermore, encapsulation of curcumin in the NL formulations led to significant neuroprotection against the Aβ-induced toxicity in vitro. Interestingly, this was the case for NLC and NLPC in terms of cell viability measured by MTT, while this was noted for NLPC and NLPPC when analyzing the number of apoptotic nuclei. MTT is a measure of metabolic activity, which could indicate cell stress in the absence of cell death, while quantitation of fragmented nuclei is a clear indicator of cell death. Although NLPPC neuroprotection was not clearly shown using MTT, the absence of apoptotic nuclei is a promising indicator of the neuroprotective effect of NLPPC.

We previously showed that NL particles can be internalized by neurons through a process resembling endocytosis [13]. We would thus propose that upon exposure to neurons, the curcumin-loaded NL formulations were internalized, allowing direct access to curcumin intracellularly. This explains the lack of cytotoxicity of curcumin when encapsulated in NLC (Figure 3), as well as the beneficial neuroprotective properties that were observed (Figure 4 and Figure 5).

These formulations retained or improved bioactivity and presented significant advantages by decreasing the intrinsic cytotoxicity of curcumin. NLP and NLPP should allow the administration of higher doses of bioactive hydrophobic curcumin, thereby delivering an effective dose to the brain after oral administration. These results demonstrate the feasibility of using RGD peptide-PEGylated NL (NLPP) for encapsulation of curcumin as a potential means for administering PUFA under optimal conditions.

## 4. Conclusions

To conclude, this study examined the potential use of curcumin as a neuroprotective agent when enclosed within peptide-PEGylated nanoliposomes prepared using PUFA-rich salmon-derived lecithin. Curcumin is known for its anti-inflammatory, neuroprotective, and antioxidant attributes; nevertheless, applications are limited by its sensitivity to environmental factors and its limited water solubility. Our findings demonstrated that the formulations of RGD peptide-PEGylated PUFA-rich lecithin liposomes were spherical, negatively charged, and sized in the nanoscale range. In cytotoxicity experiments, neurons exhibited an improved ability to tolerate curcumin when administered in the form of liposomes, as compared to free curcumin. Furthermore, in a model of Aβ-induced toxicity, cells pre-treated with curcumin-loaded nanoliposomes displayed markedly reduced cellular toxicity compared to control groups. These outcomes underscore the promise of utilizing nanoliposomes derived from salmon lecithin as an efficient carrier for delivering curcumin as a neuroprotective agent. The BBB permeability of curcumin in the different NL formulations (i.e., empty NL and curcumin-loaded NL either with or without PEG and peptide) is currently under investigation. Nevertheless, we propose that this approach overcomes the challenges associated with curcumin’s inherent properties, and represents a strong potential for applications in developing strategies of neuroprotection.

## Figures and Tables

**Figure 1 pharmaceutics-15-02665-f001:**
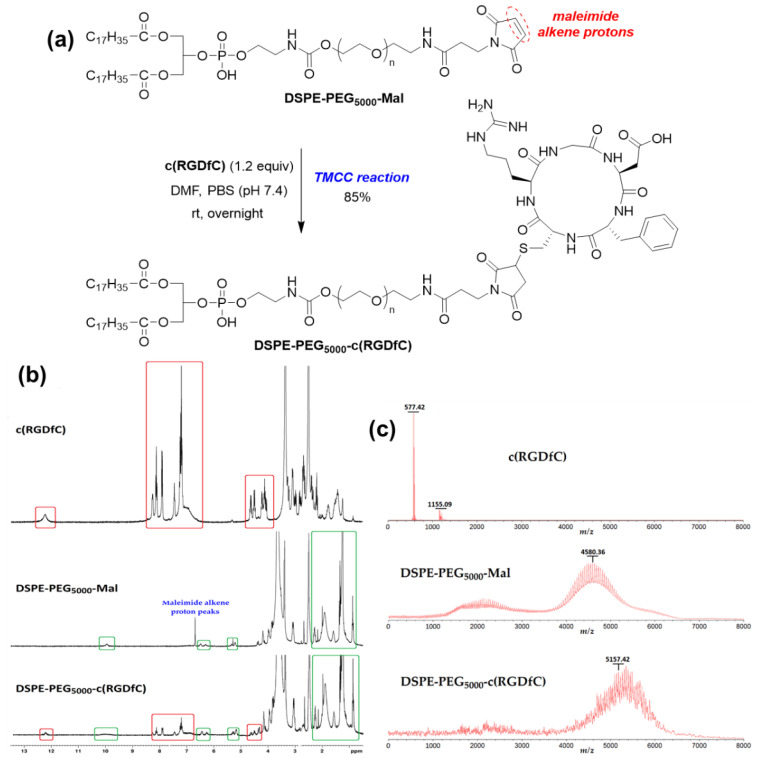
(**a**) Synthesis of DSPE-PEG_5000_-c(RGDfC). (**b**) ^1^H NMR spectra (300 MHz, DMSO-*d*_6_) of c(RGDfC), DSPE-PEG_5000_-Mal, and DSPE-PEG_5000_-c(RGDfC). (**c**) MALDI-TOF mass spectra of c(RGDfC), DSPE-PEG_5000_-Mal, and DSPE-PEG_5000_-c(RGDfC).

**Figure 2 pharmaceutics-15-02665-f002:**
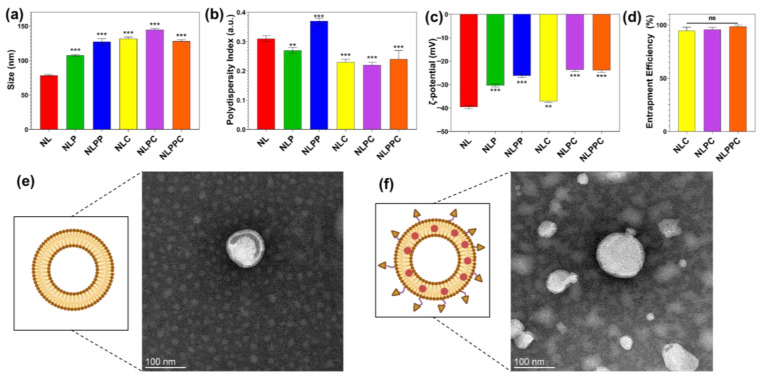
Physicochemical characterizations of empty NL and curcumin-loaded NL with or without PEG coating (NLPC and NLC, respectively), and peptide coating (NLPPC). (**a**) Size, (**b**) polydispersity, (**c**) ζ-potential, and (**d**) encapsulation efficiencies of NLC, NLPC, and NLPPC. Schematic and TEM images showing (**e**) NL and (**f**) NLPPC spherical morphologies. Created with BioRender.com. All data are expressed as mean ± standard deviation. Significance is indicated as ns (not significant), ** (*p* < 0.01), and *** (*p* < 0.001).

**Figure 3 pharmaceutics-15-02665-f003:**
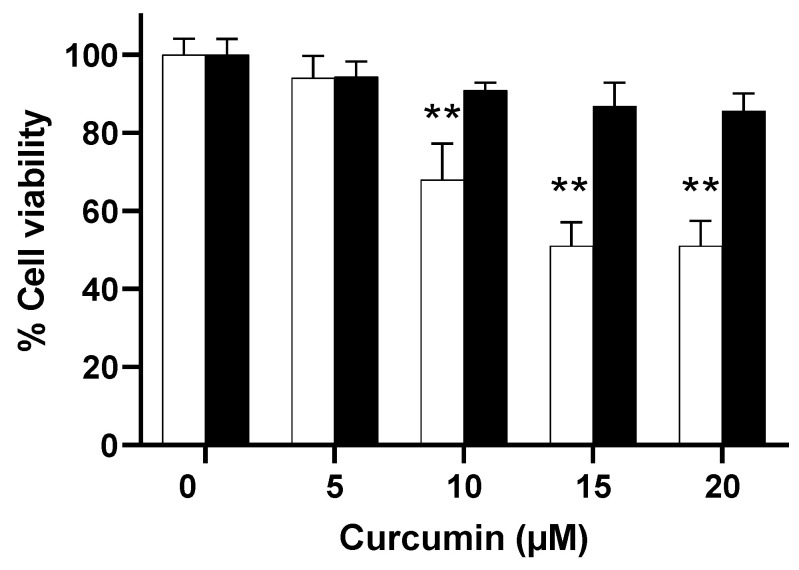
Effect of curcumin encapsulation in NL on cell viability. Primary cultures of rat embryo cortical neurons were incubated with free curcumin (☐) or curcumin loaded in 10 µg/mL NLC at the indicated concentrations (■). Results are shown as mean ± SD (*n* = 4); ** *p* < 0.01, as compared to the corresponding control (in the absence of curcumin).

**Figure 4 pharmaceutics-15-02665-f004:**
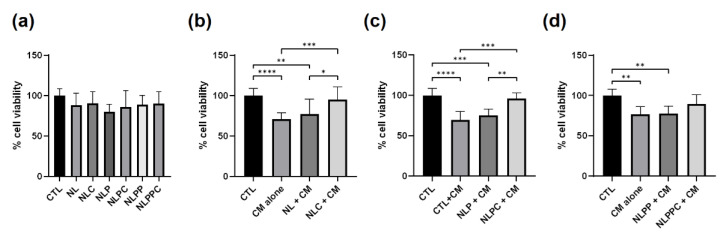
Tests of cytotoxicity (**a**) and neuroprotection (**b**–**d**) of empty and curcumin-encapsulated NL formulations on differentiated SH-SY5Y WT neuronal cell line. (**a**) Differentiated SH-SY5Y WT neurons were incubated for 24 h at 37 °C with 10 µg/mL empty NL, NLP, NLPP, or 10 µg/mL of curcumin (20 µM) loaded NLC, NLPC, or NLPPC, followed by an analysis of cell viability using MTT. (**b**–**d**) In another set of cells, differentiated SH-SY5Y WT neurons were pre-incubated for 24 h at 37 °C with the different NL formulations ((**b**) NL, NLC; (**c**) NLP, NLPC; (**d**) NLPP, NLPPC) and then exposed to conditioned media (CM) from SH-SY5Y APP cells containing Aβ peptide. Since each formulation was tested on a different plate, a set of controls was performed for each NL formulation. Cell viability was determined using MTT. Results are shown as % of cell viability compared to the corresponding controls (mean ± SD; N = 3, *n* = 3) (*, *p* < 0.05, **, *p* < 0.01, ***, *p* < 0.001, ****, *p* < 0.0001).

**Figure 5 pharmaceutics-15-02665-f005:**
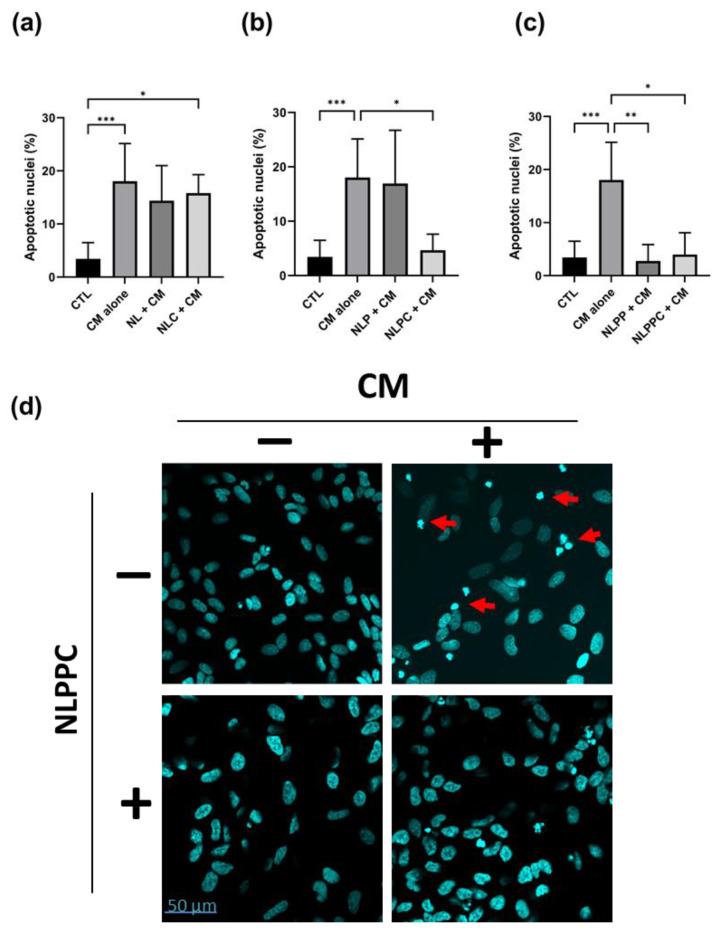
Neuroprotection of empty and curcumin-encapsulated NL formulations on differentiated SH-SY5Y WT neuronal cell line. Differentiated SH-SY5Y cells on slides were incubated for 24 h at 37 °C with 10 µg/mL empty (NL, NLP, or NLPP) or curcumin-loaded (NLC, NLPC, or NLPPC) formulations, followed by incubation for 24 h at 37 °C in the presence of CM containing Aβ peptide obtained from SH-SY5Y APP cells. Cells were washed, and fixed, followed by staining with DAPI and visualized using confocal microscopy to quantitate apoptotic nuclei. Results are shown for each formulation type ((**a**) NL, NLC; (**b**) NLP, NLPC; (**c**) NLPP, NLPPC). Representative images are shown in (**d**) for cells incubated in the absence and presence of CM or NLPPC, as indicated (fragmented nuclei are indicated with a red arrow). Results are shown as the mean number of apoptotic nuclei ± SD counted in *n* = 5 fields. (* *p* < 0.05, ** *p* < 0.01, *** *p* < 0.001; the control (CTL) and CM alone are identical and shown on each graph to indicate the comparisons of statistical significance).

## Data Availability

All data are contained within the paper.

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
