# Peer review of "Neuroprotective Effect of Curcumin-Loaded RGD Peptide-PEGylated Nanoliposomes"

_pharmaceutics, 2023, doi:10.3390/pharmaceutics15122665_

Round 1

Reviewer 1 Report

Comments and Suggestions for Authors

1. English has to be thoroughly proofread and corrected - plenty of infinitives (generally, they are accepted in materials and methods section); narrative has to be unified - generally past simple for obtained results, and present simple for know facts, some intended exceptions are allowed;

2. Introduction - the logical transition along study has to be clarified: AD is a global threat > PUFA are important, but have to be delivered properly > earlier we formulated PUFA-rich NL based on salmon lecithin > we optimized delivery by DSPE-PEG-cRGD incorporation into NL > now we supplemented these NL with curcimin > in vitro studies revealed promises. 

3. Mistyping errors - line 53 (red dot), 72, etc.

4. fig 5 - hash and tab signs make the conclusion much more obscure. Is it possible to clarify the figure? Because now I see following things from the figure: first - 1st column a/b/c, CM decreases survival of CTL; second - 2nd column a/b/c, NL/NLP/NLPP did not increase significantly survival comparing with 1 column (CTL + CM), but at the same time the formulations itself decrease survival; third - 3rd column a/b/c,  NLC/NLPPC did not increase significantly survival comparing with 1 column (CTL + CM), only in NLPC case, according SD I see significant difference, but at the same time the formulations itself decrease survival (non significantly). I propose to think how to represent these results in a more obvious way. Is there any hope that incubation time will influence the survival gap among the samples?

5.  The short rationale of the surface cRGD peptide application is missed in the introduction and results sections. And general issue - I understand integrins targeting or blockade through cRGD peptide, but its application in this specific study is obscure - lack of binding or targeted transport studies are performed, but at the same time the DSPE-PEG-cRGD synthesis took the significant manuscript place. It seems illogical, but maybe I have missed something.

6. Animal intervention: primary culture of rat embryos cortical neurons - ethical committee approval protocol number and date are missing. 

7. It would be great if the authors provide primary data from Malvern ZetaSizer Data in supplementary materials: size distribution with polydispersity indexes and surface charge graphs.

8. Since, the live cell membranes are also permeable for DAPI - it would be great if the authors can provide fluorescent images of the cells treated with the liposomes along with the bars at Fig. 6. The protocol of DAPI nuclei staining should be clarified (time, concentrations) or supplemented with appropriate reference. 

9. Curcumin water solubility is about 1.6 mkM/L. Thus, curcumin solubilization process and solvent should be clarified, as well as influence of the solvent on cell survival (fig. 4).

10. The Aβ peptides presence in CM should be proved or appropriate reference provided in the methods section.  

Summarizing, I sympathize with the general idea of the article, but the mentioned issues should be addressed. 

Comments on the Quality of English Language

English has to be thoroughly proofread and corrected

Author Response

  1. English has to be thoroughly proofread and corrected - plenty of infinitives (generally, they are accepted in materials and methods section); narrative has to be unified - generally past simple for obtained results, and present simple for known facts, some intended exceptions are allowed;

    - Response:
    We thank the reviewer for this comment. We have proof-read the manuscript and corrected the mistakes along the manuscript. Changes are marked in green.
  2. Introduction - the logical transition along study has to be clarified: AD is a global threat > PUFA are important, but have to be delivered properly > earlier we formulated PUFA-rich NL based on salmon lecithin > we optimized delivery by DSPE-PEG-cRGD incorporation into NL > now we supplemented these NL with curcimin > in vitro studies revealed promises.

    - Response:
    We thank the reviewer for this comment. We have re-written the introduction to make the logical transition along study clearer to the reader. Changes are marked in green.
  3. Mistyping errors - line 53 (red dot), 72, etc.

    - Response: We thank the reviewer for these observations. We have amended all mistyping errors in the manuscripts. Changes are marked in green.
  4. fig 5 - hash and tab signs make the conclusion much more obscure. Is it possible to clarify the figure? Because now I see following things from the figure: first - 1st column a/b/c, CM decreases survival of CTL; second - 2nd column a/b/c, NL/NLP/NLPP did not increase significantly survival comparing with 1 column (CTL + CM), but at the same time the formulations itself decrease survival; third - 3rd column a/b/c,  NLC/NLPPC did not increase significantly survival comparing with 1 column (CTL + CM), only in NLPC case, according SD I see significant difference, but at the same time the formulations itself decrease survival (non significantly). I propose to think how to represent these results in a more obvious way. Is there any hope that incubation time will influence the survival gap among the samples?

- Response: We have modified both Figures 4 (cell viability) and 5 (apoptotic nuclei) (formerly Figures 5 and 6) to clarify the presentation of the results, as well as the corresponding text and figure legends.  For both figures, it was necessary to keep the graphs separate to clearly show the statistical significance.
We used 24h pretreatment based on our previous studies showing internalization of the NL (Passeri et al, 2022, https://doi.org/10.3390/pharmaceutics14102172). In addition, this is sufficient time for incorporation of fatty acids from NL shown by increased cell PUFA content (Passeri et al, 2021, https://doi.org/10.3390/ijms222111859).

  1. The short rationale of the surface cRGD peptide application is missed in the introduction and results sections. And general issue - I understand integrins targeting or blockade through cRGD peptide, but its application in this specific study is obscure - lack of binding or targeted transport studies are performed, but at the same time the DSPE-PEG-cRGD synthesis took the significant manuscript place. It seems illogical, but maybe I have missed something.

    - Response:
    We thank the reviewer for this comment. We have added a section marked in green in the introduction to describe the rationale behind the use of RGD peptide in this paper.
  2. Animal intervention: primary culture of rat embryos cortical neurons - ethical committee approval protocol number and date are missing.

    - Response: B54-547-24 (Nov 27, 2015, DDPP, Meurthe-et-Moselle).
  3. It would be great if the authors provide primary data from Malvern ZetaSizer Data in supplementary materials: size distribution with polydispersity indexes and surface charge graphs.

- Response: the primary data presented in graph or in value, we choose the value presented in fig 2(a,b,c)

  1. Since, the live cell membranes are also permeable for DAPI - it would be great if the authors can provide fluorescent images of the cells treated with the liposomes along with the bars at Fig. 6. The protocol of DAPI nuclei staining should be clarified (time, concentrations) or supplemented with appropriate reference.

    - Response: We have provided representative images used to assess the pro-apoptotic effect of CM. The protocol has been added in the Materials and Methods section.
  2. Curcumin water solubility is about 1.6 mkM/L. Thus, curcumin solubilization process and solvent should be clarified, as well as influence of the solvent on cell survival (fig. 4).

    - Response: We thank the reviewer for this comment, but we think there was some misunderstanding. In the section “4.2.3 Preparation of nanoliposomes” we mention that curcumin and lecithin were dissolved in a mixture of chloroform:methanol and not in water. Following that, a thin lipid film was formed by completely evaporating the solvent under vacuum at 50 °C. So, the organic solvents are completely evaporated and are never in contact with cells.
  3. The Aβ peptides presence in CM should be proved or appropriate reference provided in the methods section.

- Response:   The Aβ peptides levels in the CM were measured using an immunoassay: Aβ 1-40: 568 ± 120.2 pg/mL and Aβ 1-42 41.3 ± 13.6 pg/mL.  This information has been included in the text and the methods sections marked in green.

Summarizing, I sympathize with the general idea of the article, but the mentioned issues should be addressed.

- Response: We thank the reviewer for his/her helpful comments that significantly improved our manuscript. We hope that our corrections answered all his/her concerns.

Comments on the Quality of English Language

English has to be thoroughly proofread and corrected

Reviewer 2 Report

Comments and Suggestions for Authors

In the article titled "Neuroprotective effect of curcumin-loaded peptide-targeted PEGylated liposomes" the authors have evaluated the potential application of curcumin as a neuroprotective agent encapsulated in peptide-targeted PEGylated liposomes developed from marine-derived lecithin. This paper can be considered for publication after addressing following concerns: 

1.Clarity and Conciseness: While the introduction provides a comprehensive overview of the research topic, it could benefit from greater conciseness. Some sentences are quite long and complex, which may make it harder for readers to grasp the main points.

2. Citations: While the introduction includes some citations to previous research (e.g., [1], [3], [4], [6]), it would be helpful to include more specific information about these references, such as the authors' names and the publication years, to provide context and credibility to the claims made.

3. In the introduction the authors need to emphasis what is the current knowledge gap and how this article addresses that?

4. Provide Figure 1 as a supporting information.

5. How is it possible for NLPPC to have entrapment efficiency of 98.4±1.4 %?

6.Figure 3 e and f the schematic provided does not match with the TEM images, the TEM images and non-conclusive. Provide better images where the distinction is visible. 

7. Figure 5 and 6 are hard to understand, they need to be presented in a better and more easy understand way.

Author Response

In the article titled "Neuroprotective effect of curcumin-loaded peptide-targeted PEGylated liposomes" the authors have evaluated the potential application of curcumin as a neuroprotective agent encapsulated in peptide-targeted PEGylated liposomes developed from marine-derived lecithin. This paper can be considered for publication after addressing following concerns: 

1.Clarity and Conciseness: While the introduction provides a comprehensive overview of the research topic, it could benefit from greater conciseness. Some sentences are quite long and complex, which may make it harder for readers to grasp the main points.

- Response: We thank the reviewer for this comment. We have re-written, added, and deleted some sections in the introduction to make it clearer to the reader. Changes are marked in green.

  1. Citations: While the introduction includes some citations to previous research (e.g., [1], [3], [4], [6]), it would be helpful to include more specific information about these references, such as the authors' names and the publication years, to provide context and credibility to the claims made.

- Response:
We thank the reviewer for this comment. We have added author names to refs [1] and [4]. Changes are marked in green.

  1. In the introduction the authors need to emphasis what is the current knowledge gap and how this article addresses that?

- Response:
We thank the reviewer for this comment. We have re-written the introduction to make the logical transition along study clearer to the reader. The logical transition along study was clarified to be: AD is a global threat > PUFA are important, but have to be delivered properly > earlier we formulated PUFA-rich NL based on salmon lecithin > we optimized delivery by DSPE-PEG-cRGD incorporation into NL > now we supplemented these NL with curcumin > in vitro studies revealed promises. Changes are marked in green.

  1. Provide Figure 1 as a supporting information.

- Response:
We thank the reviewer for this suggestion. We have moved Figure 1 to supporting information section.

  1. How is it possible for NLPPC to have entrapment efficiency of 98.4±1.4 %?

- Response:
Curcumin is highly soluble in polar solvents such as methanol. In this study we solubilized curcumin in methanol then encapsulated it in liposomes using the thin film method. Since it is a hydrophobic molecule it will be entrapped in the lipid membrane which has a large surface area. For this reason, we achieved a high entrapment efficiency. If it was a hydrophilic molecule the entrapment efficiency would be much lower because the core area of liposomes is much smaller than the lipid membrane area.

  1. Figure 3 e and f the schematic provided does not match with the TEM images, the TEM images and non-conclusive. Provide better images where the distinction is visible. 

    - Response:
    We thank the reviewer for this comment. We have edited the schematics by decreasing the size of PEG-peptide to be closer to reality. However, we disagree that TEM images are non-conclusive, since our purpose was to observe the spherical shape of nanoliposome formulations which is clearly shown in the TEM images.

  1. Figure 5 and 6 are hard to understand, they need to be presented in a better and more easy understand way.

- Response: Our apologies for the confusion, we have presented Figures 5 and 6 differently (now Figures 4 and 5), to be more clear.  As we indicated in our answer to Referee 1, we have modified both figures to clarify the presentation of the results, as well as the corresponding text and figure legends.  For both figures, it was necessary to keep the graphs separate to clearly show the statistical significance.

Reviewer 3 Report

Comments and Suggestions for Authors

In this research article, the authors examined the potential neuroprotective effect of curcumin by encapsulating the compound into fabricating nanoliposomes. Drug encapsulation has been a promising research area of high interest for many years. However, this paper needs substantial improvements to provide evidence and other necessary information about their findings. There are also inconsistencies in the text. Some of the examples can be as follows.

1.     What do the authors mean by suggesting ‘preventive approaches to slow down the physiological ageing of the brain’ (Introduction)? How does it relate to applying liposomes, especially as it has been rightly pointed out that ‘the irreversible damage has already occurred by the time the first clinical signs appear’?

2.     ‘Brain-targeting of nanoliposomes’ appears unclear in the context of their delivery through blood circulation (lines 81-87). The authors should describe more findings from the literature, with attempts to deliver nanoliposomes to the brain.

3.     There is a general lack of discussion and in-depth description of findings as well as comparing them with those of others.

Major issues about experimental findings:

 -       Why does  curcumin produce harmful effects in neurons (Fig. 3)? The authors have to describe and discuss, at least in brief, the mechanism of curcumin action in neurons.

-       On what basis did the authors compare the effects of curcumin alone and encapsulated one (Fig. 3)? Did they consider the effectiveness rate for curcumin encapsulation? How do the authors know the concentration of curcumin in nanoliposomes? How many liposomes did they add to neurons to directly compare concentrations? Please explain.

-       How encapsulated curcumin could act on neurons? Do neurons uptake nanoliposomes? If yes, it should be shown here.

-       What was the concentration (amount) of the Aβ peptide in conditioned media?

-       Why did the authors use SH-SY5Y neuronal line but not primary cortical neurons, on which they tested the effect of curcumin’s free and encapsulated compound?

-       In Figure 6, please add some representative images to demonstrate apoptotic nuclei.

Comments on the Quality of English Language

Extensive editing of English language required

Author Response

  1. What do the authors mean by suggesting ‘preventive approaches to slow down the physiological ageing of the brain’ (Introduction)? How does it relate to applying liposomes, especially as it has been rightly pointed out that ‘the irreversible damage has already occurred by the time the first clinical signs appear’?

- Response: Previous literature has shown the neuroprotective aspects of omega-3 fatty acids that should favor resistance of neurons to limit age-related cognitive decline. These PUFA-rich NL could provide such a means towards this end. We have modified the text to clarify this. 

  1. ‘Brain-targeting of nanoliposomes’ appears unclear in the context of their delivery through blood circulation (lines 81-87). The authors should describe more findings from the literature, with attempts to deliver nanoliposomes to the brain. 

- Response: We previously reported that oral administration of empty NL led to PUFA-enrichment in the brain, suggesting brain bioavailability of NL. A discussion on brain-targeted delivery of NL has been added to the introduction. Changes are marked in green.

  1. There is a general lack of discussion and in-depth description of findings as well as comparing them with those of others.

 Major issues about experimental findings:

 -       Why does curcumin produce harmful effects in neurons (Fig. 3)? The authors have to describe and discuss, at least in brief, the mechanism of curcumin action in neurons.

- Response: In most articles on the neuroprotective effects of curcumin, a cell viability dose-response is presented to validate the highest non-toxic dose to be used. Several references were added in the text to illustrate the results usually obtained. The neurotoxic concentration varies according to the model used and must be assessed in each study, as we have done here. The mechanisms involved have not been described.

-       On what basis did the authors compare the effects of curcumin alone and encapsulated one (Fig. 3)? Did they consider the effectiveness rate for curcumin encapsulation? How do the authors know the concentration of curcumin in nanoliposomes? How many liposomes did they add to neurons to directly compare concentrations? Please explain. 

Response: Figure 2 shows that encapsulation efficiency is similar for all NL formulations, between 94 and 98%, with no significant difference between all 3 formulations.  We added similar amounts based on the amount of salmon lecithin of each NL formulation, both empty and curcumin-loaded. We have included a statement to clarify this in the text marked in green.

-       How encapsulated curcumin could act on neurons? Do neurons uptake nanoliposomes? If yes, it should be shown here. 

Response: We have previously shown that NL particles can be internalized by neurons through an endocytic process. Because of this, we would expect the curcumin to be internalized, allowing direct access of curcumin intracellularly. This would explain the lack of neurotoxicity when curcumin is encapsulated with NL. We have added this in the discussion marked in green.

-       What was the concentration (amount) of the Aβ peptide in conditioned media? 

Response: The Aβ peptides levels in the CM were verified by immunoassay (automat Lumipulse G120, Fujirebio, Courtaboeuf, France), and found to be 568 ± 120.2 pg/mL (Aβ 1-40) and 41.3 ± 13.6 pg/mL (Aβ 1-42)(n = 6).  This information has been included in the text and the methods section.

-       Why did the authors use SH-SY5Y neuronal line but not primary cortical neurons, on which they tested the effect of curcumin’s free and encapsulated compound? 

Response: We found comparable results between primary cultures of rat embryo cortical neurons (ref 22, Colin et al, 2016) and the SH-SY5Y neurons with regards to the effects of CM. We chose to use the cell line to minimize use of protocols requiring animals, thereby applying the 3R rule (replacement, reduction, and refinement) in the use of animals for experimentation.  

-       In Figure 6, please add some representative images to demonstrate apoptotic nuclei.

Response: We have included some representative images of the apoptotic nuclei in Figure 5 (Previously 6).

Round 2

Reviewer 1 Report

Comments and Suggestions for Authors

The authors have to include animal committee approval and date of approval they provided earlier in corresponding section in materials and methods.

The other issues I mentioned have been addressed.

Author Response

we included ,the authors have to include animal committee approval and date of approval they provided earlier in corresponding section in materials and methods.

Reviewer 2 Report

Comments and Suggestions for Authors

The manuscript can be accepted in its current form after authors have the required major revision 

Author Response

the English was revised by the American researchers 

Reviewer 3 Report

Comments and Suggestions for Authors

Figure 5D should be enlarged to visualise single cells.

Comments on the Quality of English Language

The paper has been improved but not fully. 

Author Response

We improved the quality of fig 5